# *Trichoderma asperellum* L. Coupled the Effects of Biochar to Enhance the Growth and Physiology of Contrasting Maize Cultivars under Copper and Nickel Stresses

**DOI:** 10.3390/plants12040958

**Published:** 2023-02-20

**Authors:** Fatima Amanullah, Waqas-ud-Din Khan

**Affiliations:** 1Sustainable Development Study Centre, Government College University, Lahore 54000, Pakistan; 2Department of Agriculture, Government College University, Lahore 54000, Pakistan; 3Tasmanian Institute of Agriculture, University of Tasmania, Hobart 7005, Australia

**Keywords:** *Trichoderma harzianum*, *Trichoderma viride*, biosorption, antioxidants, heavy metals

## Abstract

Crop cultivation in heavy metal (HM)-polluted soils is a routine practice in developing countries that causes multiple human health consequences. Hence, two independent studies have been performed to investigate the efficiency of rice husk biochar (BC) and three fungal species, *Trichoderma harzianum* (F1), *Trichoderma asperellum* (F2) and *Trichoderma viride* (F3), to improve the growth and physiology of *Zea mays* L. plants grown on soil contaminated with Cu and Ni. Initially, a biosorption trial was conducted to test the HM removal efficiency of species F1, F2 and F3. Among them, F2 sp. showed the maximum Cu and Ni removal efficiency. Then, a pot study was conducted with two cultivars (spring corn and footer corn) having eleven treatments with three replicates. The results demonstrated a significant genotypic variation among both cultivars under applied HM stress. The maximum decreases in leaf Chl a. (53%), Chl b. (84%) and protein (63%) were reported in footer corn with applied Cu stress. The combined application of biochar and F2 increased leaf CAT (96%) in spring corn relative to Cu stress. Altogether, it was found that BC + F2 treatment showed the maximum efficiency in combatting Cu and Ni stress in spring corn.

## 1. Introduction

Due to rapid industrialization and poor recycling mechanisms in developing countries, heavy metals are the major contaminants reaching the agricultural fields; hence, the quality of food is getting compromised. For example, the accumulation of copper (Cu) can reach up to ~1170 mg·kg^−1^ and ~1500 mg·kg^−1^ in sludge-treated and in top vineyard soil [1]. Plants show various Cu and nickel (Ni) toxicity symptoms, such as retardation in root and shoot growth, reduction in plant weight and alteration in germination processes [2]. These metals also cause oxidative stress in plants, as there is surplus production and concentration of reactive oxygen species (ROS) suppressing the natural homeostasis and balance of plant antioxidant enzymes and secondary metabolites [2]. Hence, a significant loss in crop yield and its productivity have been reported by different scientists [3,4].

Researchers concluded that the previous technologies used to mitigate the Cu and Ni toxic effects have various limiting factors including excessive cost, complexity of procedure, cause of secondary pollution, and modification of the physical and chemical nature of the environment [5,6]. In addition, these techniques subsequently remove non-target beneficial microbiota such as nitrogen-fixing bacteria as well as other fauna species [7]. To combat those limitations, biological methods are highly recommended because they are eco-friendly, fast, and cost-effective [8,9]. In addition, the use of biological methods allows the treatment of a large volume of effluent with minute inoculum concentrations and short operation times [10]. Biosorption technique is prevalent in environmental science where microbes are used to detoxify the HMs [7]. Among the microbes, fungi are supposed to have the better HMs sorption capacity not only due to their physical and biological nature but also their ability to accumulate the HMs intracellularly through their living biomass, and/or by binding both living and dead biomass extracellularly [11,12,13]. It is known that *Trichoderma* species have the ability to produce various kinds of enzymes to degrade substrates. The principal uses of the fungi such as *T. harzianum* to reduce HMs are either through biosorption, bioaccumulation, bioprecipitation, bioreduction, and/or bioleaching [14]. A study reported that tested fungal isolate *T. lixii* (CR700) @ 10 mg·L^−1^ significantly removed As (95%) and Ni (68%) [15]. This is due to the surface sorption ability of *T. lixii* where glutathione involved in the detoxification of xenobiotics and thiol-based antioxidants played a significant role. Similarly, researchers have analysed the comparative Cd removal efficiency of six *Trichoderma sp*. and reported that *T. simmonsii* (UTFC 10063) are tolerant to Cd toxicity and showed up to 92 and 32% removal efficiency at 10 and 500 mg·L^−1^, respectively [16].

Similarly, the utilization of biochar is commonly believed to be an efficient amendment that can reduce the mobility and availability of these metals in polluted soil, hence reducing their accumulation in plants [17,18]. Bioremediation through the value addition processes such as biochar/engineered biochar are recent methods used to mitigate the harmful effects of lead (Pb) and chromium (Cr) toxicity in the soil [19,20]. The *T. harzianum* sp. and sewage sludge biochar was shown to give a significant improvement in soybean dry biomass (70%) and seed germination (20%) relative to control as reported by [21]. The results of Herliana [22] showed that the adsorption of Pb was enhanced in soil under the application of organic adsorbents (biochar and *T. harzianum*).

Similarly, genotypic variation in crops and their morphological/physiological activity against HM uptake is a good criterion to evaluate the dose response. A study investigated five cultivars of maize and revealed that the genotype *31R88* was a more sensitive cultivar, while Neelum was a tolerant maize cultivar against Cr and Mn stress [23]. In the study of [24], the sorghum cultivar JS-2002 has shown a higher Cd tolerance potential due to its efficient antioxidant defense system than the sensitive Chakwal Sorghum. Similarly, four maize cultivars (NK-8441, P-1543, NK-8711, and FH-985) were grown in the Cr-polluted soils with 3% of sugarcane bagasse biochar [25]. The study concluded that cultivar P-1543 performed well under both Kasur and Sialkot Cr-polluted soil by reducing plant cell electron leakage and uptake of Cr in aerial parts of plant and increased the relative water content, plant dry biomass and antioxidant enzyme production. The results were consistent with the study of [26], who investigated the Cd-tolerance ability of two cultivars (hybrid and late japonica) of *Oryza sativa* L. with shell-derived biochar (KC) or maize straw-derived biochar (MC). They reported that biochar significantly reduces bioaccumulation of Cd; however, the hybrid rice cultivar showed a higher bioaccumulation factor than japonica rice and should not be cultivated. Similarly, it is reported that the maize cultivar-6103 remained as Cr-tolerant while the cultivar-9108 was sensitive to Cr stress on the basis of their biochemical parameters [27]. Likewise, the four barley genotypes showed different levels of Cd stress tolerance in terms of plant growth, biomass, chlorophyll content, lipid peroxidation and antioxidant activities [28].

Recently, we investigated the potential of *Trichoderma* sp. with iron and zinc (Zn)-enriched biochars against soil Cr and Pb immobilization [19]; however, a detailed study was required to categorize the different fungal species on their HM removal efficiency and then select the most suitable fungal sp. to evaluate its effects with/or without biochar on different maize cultivars. It was hypothesized that the synergistic effect of BC and *Trichoderma* sp. might efficiently enhance the physiological and biochemical parameters of the *Zea mays* L. plant by regulating the effects of HM stress in soil. More specifically, we tend to screen the *Trichoderma* isolates on the basis of their removal efficiency against HMs (Cu, Ni and Pb); then, examining the effects of *Trichoderma asperellum* with BC on physiological and biochemical parameters of two cultivars against selected HMs; and, lastly, assess the genotypic variation of maize cultivars on the basis of their HM tolerance.

## 2. Results

### 2.1. Cu, Ni and Pb Removal through Fungal Strains in Aqueous Media

The two fungal strains *T. harzianum* (F1) and *T. asperellum* (F2) showed relatively high Cu content in fungi dry biomass as compared to *T. viride* (F3) (Figure 1). Out of three HM-resistant isolates at day 5, F2 showed the maximum Cu content @ 300 ppm up to (38%) and (1200-fold) when compared with F1 and F3, respectively (Figure 1). The investigation found that F2 showed the most effective sorption (200 and 2-fold) of total Ni @ 150 ppm when compared with F1 and F3, respectively (Figure 1).

All fungal strains demonstrated relatively high total Cu and Ni tolerance (Figure 2). Out of three fungal isolates at day 5, F2 showed the maximum total Cu removal efficiency (11 and 9%) @ 300 ppm when compared with F1 and F3, respectively. Similarly, the removal efficiency of F2 was found to be 73 and 10% higher as compared to F1 and F3 with Cu @ 300 ppm at day 7, respectively. It was observed that the removal efficiency of *T. asperellum* (F2) remained consistent under applied Ni and Pb stress. The maximum HM sorption capacity under Ni stress at day 7 was demonstrated by F2 @ 150 ppm up to 7 and 1% as compared to F1 and F3, respectively (Figure 2).

It was reported that the HM adsorbed per DW of fungal isolate F2 was relatively higher at day 7 as compared to F1 and F3 (Figure 3). Under Cu toxicity, there was an increase (13 and 23%) in Cu uptake per DW of fungi reported with F2 application as compared to F1 and F3, respectively. Similarly, Ni uptake of *T. asperellum* (F2) was 9-fold higher as compared to F3 @ 150 ppm. However, the Pb uptake by F2 showed a negative trend at day 5 with respect to F1 and F2, while F2 demonstrated the maximum Pb uptake per DW (10- and 5-fold) of fungi at day 7 as compared to F1 and F3 with Pb @ 150 ppm, respectively. 

### 2.2. Effect of T. asperellum and Biochar on Growth and Physiology of Maize Cultivars

The plant length, fresh and dry weight were also significantly (*p* ≤ 0.05) influenced by genotypic variations (Table 1). The root length of spring corn is significantly reduced (46 and 60%) under Cu- and Ni-contaminated soil as compared to control plants. The maximum decrease (45%) in shoot FW was reported in footer corn with Ni stress compared to control. However, *T. asperellum* and BC-combined application enhanced the shoot and root FW under Cu toxicity irrespective of the cultivars. The maximum increase (1-fold) in shoot FW was reported in spring corn with treatment (BC + *T. asperellum* +Cu) relative to Cu stress. Similarly, the dry weight (DW) of maize plant for both cultivars also showed the similar trend with the maximum shoot DW up to (2-fold) was shown by spring corn in treatment (BC + *T. asperellum* + Cu) as compared with Cu stress. In case of Ni toxicity, both FW (36%) and DW (77%) of maize shoots were also significantly reduced in spring corn as compared to control plants.

Similarly, leaf chl. a was significantly reduced (75%) under Ni toxicity relative to control in spring corn (Figure 4). However, *T. asperellum* and BC-combined application enhanced the leaf chl. a and b under Cu toxicity irrespective of the cultivars. The maximum increase (51%) in leaf chl. a was reported in spring corn with treatment (BC + *T. asperellum* + Cu) relative to Cu stress. Similarly, the maximum leaf phenolics (2-fold) was also reported in spring corn by BC + *T. asperellum* + Cu as compared with control. In case of Ni toxicity, the combined application of *T. asperellum* and BC increased leaf phenolics (16%) and root phenolics (21%) in spring corn relative to applied Ni stress. Similarly, applied Cu and Ni stress significantly (*p* ≤ 0.01) reduced the spring corn total protein concentration (Figure 4). The maximum decrease (64%) in leaf protein was reported in footer corn with Cu stress. Under BC + *T. asperellum* + Cu treatment, plants showed an increase (50%) in leaf protein concentration as compared to Cu in spring corn. Similarly, under BC + *T. asperellum* + Ni treatment, plants showed an increase (40%) in leaf protein concentration as compared to Ni in spring corn.

It was also examined whether roots and aerial parts of both cultivars exhibited elevated concentrations of Cu and Ni when exposed to high concentrations of Cu and Ni @ 300 and 100 mg Kg^−1^, respectively (Figure 5), which revealed that both cultivars accumulated the Cu and Ni in its different sections such as root, stem and leaves. However, *T. asperellum* and BC application significantly regulated the effect of total Cu and Ni in plants. The effect of *T. asperellum* was analysed as a potential HM stress-tolerant sp. which significantly reduced the uptake of total Cu (71%) and Ni (30%) in spring corn leaf; however, *T. asperellum* + BC (T10) combined application showed the most efficient results to regulate total Cu (83%) and Ni (66%) conc. in leaf of spring corn plant (Figure 5).

Similarly, the applied HM stresses significantly (*p* ≤ 0.01) increased the conc. of total Cu and Ni in soil (Figure 5). However, *T. asperellum* and BC application significantly regulated the effect of total Cu and Ni in soil. The effect of *T. asperellum* analysed as a potential HM stress-tolerant sp. which significantly reduced the uptake of total Cu (77%) and Ni (75%) in soil; however, *T. asperellum* + BC combined application showed the most efficient results to regulate total Cu (83%) and Ni (78%) conc. in soil (Figure 5).

The Cu and Ni stress enhanced the MDA and H_2_O_2_ contents irrespective of the cultivars (Figure 6). The maximum increases in leaf MDA (up to 6-fold) and H_2_O_2_ (76%) were reported in footer corn with applied Ni stress. However, *T. asperellum* and BC combined application decreased the leaf oxidative stress under Cu toxicity irrespective of the cultivars. The maximum decrease (86%) in leaf MDA was reported in spring corn with treatment (BC + *T. asperellum* + Cu) relative to Cu stress.

The applied HM stresses significantly inhibited the antioxidant enzymes activity in both maize cultivars (Figure 7 and Figure 8). In spring corn, leaf SOD (90%) and APX (61%) conc. was significantly reduced under Ni toxicity relative to control. However, *T. asperellum* and BC combined application enhanced the SOD and APX concentration under Cu toxicity irrespective of the cultivars. The maximum increase in leaf SOD (3-fold) and APX (50%) was reported in spring corn with treatment (BC + *T. asperellum* + Cu) relative to Cu stress (Figure 7).

Similarly, there was significant (*p* ≤ 0.05) genotypic variation in CAT and POD activity shown by both cultivars against applied stresses (Figure 8). Leaf CAT (67%) and POD (90%) conc. was significantly reduced under Ni toxicity relative to control in spring corn. However, *T. asperellum* and BC combined application enhanced the CAT and POD concentration under Cu toxicity irrespective of the cultivars. The maximum increase in leaf CAT (96%) and POD (98%) was reported in spring corn with treatment (BC + *T. asperellum* + Cu) relative to Cu stress (Figure 8).

### 2.3. Principal Component Analysis (PCA)

Physiological and biochemical parameters of both maize cultivars and soil were statistically analysed to evaluate the most efficient treatments used in the study by constructing biplots of PCA (Figure 9). PCAs contributed to 82.13% of the variance in plant and 89.48% of the variance in soil biplots (Figure 9). The results of both biplots suggested that different parameters of growth, secondary metabolites and antioxidant enzyme activity showed a highly negative correlation with applied Cu and Ni stresses, whereas HM (Cu and Ni) toxicity were highly positively correlated with concentration of ROS and accumulation of Cu and Ni in different parts of plant and in soil. In the biplot of a plant, secondary metabolites showed a highly positive relationship with antioxidant enzyme (APX, CAT, SOD and POD) activity but a highly negative relationship with ROS (H_2_O_2_ and MDA) concentrations of both maize cultivars. 

## 3. Discussion

### 3.1. Impact of Fungal Strains on Metal Removal

In this study, *T. asperellum* (F2) showed significant removal efficiency of Cu, Ni and Pb but its absurd behavior was shown in Pb uptake at day 5 and day 7 (Figure 2 and Figure 3). Similarly, the *T. asperellum* showed the maximum Cu and Ni removal efficiency with their adsorption per DW of fungal biomass (Figure 2 and Figure 3). Previous studies have reported that different *Trichoderma* sp. isolated from contaminated soil and sediments remained tolerant to Cu [29,30]. It is suggested that high Cu-tolerance in *Trichoderma* sp. might be due to the adaptation of this sp. to the polluted environment [31]. This Cu tolerance through *Trichoderma* sp. could be enhanced as a result of agricultural management practices, particularly by regulating the use of Cu-based pesticides on crops. However, Mohsenzadeh and Shahrokhi [32] reported that there is a species-specific relationship existed between fungal strain and its corresponding ability to remove HM from polluted soil. Microorganisms that adapt to and grow in HM-contaminated environments offered the mechanisms to transport the metal inside their cells and certain proteins bind it via specific enzymes and protect the microorganism from its toxic effects [29]. In case of Pb toxicity, fungal strains showed inconsistent results; hence, Pb toxicity was not taken into the pot experiment (Figure 2 and Figure 3).

### 3.2. Maize Growth and Physiology under Biochar, T. asperellum and HM Stresses 

Similarly, the growth of both maize cultivars including its fresh and dry weights were significantly (*p* ≤ 0.01) affected by the application of HM stresses (Table 1). Yang [33] suggested that a high level of Cu stress reduced the growth of root hair and damaged the root follicle, leading to severe deformations in root structure. However, BC along with F2 sp. significantly (*p* ≤ 0.01) enhanced the FW and DW in all plant parts (Table 1). It has been reported that exposure of filamentous fungi to heavy metals can lead to physiological adaptation or the selection of mutants and such changes may be associated with increased metal absorption capacity [25]. Hence, an increase in biomass of different plants (*Solanum lycopersicum*, *Beta vulgaris* and *Triticum aestivum*) were examined by the applications of different rates of biochar and *T. asperellum* under HM-stressed conditions because of their improvement in soil physio-chemical properties and an increase in availability of macro-nutrients [34,35,36,37].

Under abiotic stress, a decrease in photosynthetic pigments occur due to the degradation of enzymes (RuBP Carboxylase and ATP synthase) [38]. These enzymes are involved in the Calvin cycle and electron transport chain (ETC), and their degradation leads to damaged thylakoid membranes, chloroplast and stomatal conductance. Likewise, the chlorophyll contents (a and b) were significantly (*p* ≤ 0.01) decreased with applied stresses, especially in T2 and T3 when compared with control; however, BC application can enhance plant chlorophyll contents and its growth rate through indirectly improving the soil physiochemical parameters including water availability to the plants and improving soil natural microbial biota [19,39] (Figure 4). A study by Kumar [40] also supported such findings under Cr stress. In our previous experiments, we observed that applied biochar with *Trichoderma* sp. enhanced the porosity and surface area of soil, which increased the sorption or binding of metals and improved the functioning and synthesis of secondary metabolites such as total protein and phenolics which might be attributed to the enhanced water-holding capacity (WHC) of soil (Figure 4 and Appendix A) [19,20]. Soliman [41] also reported that the application of T. *harzianum* to *Cucurbita pepo* plants significantly improved the photosynthetic pigments and metabolites under salinity stress. Therefore, the production of secondary metabolites and photosynthetic pigments is attributed to the improved transpirational bypass flow towards leaves [42,43].

### 3.3. Maize Ionic Homeostasis under Biochar, T. asperellum and HM Stresses

Different studies have reported that application of biochar and *Trichoderma* sp. efficiently immobilizes HMs by surface adsorption due to the presence of functional groups, thus significantly decreases the uptake of Cu from soil to roots and further translocation to aerial parts of plant [44,45,46] (Figure 5). Heavy-metal-tolerant plant growth promoter (HMT-PGP) microbes such as F2 sp. alter metal bioavailability in soil through different mechanisms vis-a-vis acidification, chelation, complexation, precipitation, and redox reactions. HMT-PGP microbes are used to release different organic acids which not only lower the soil pH but also sequester soluble metal ions [46]. Experimental evidence suggests that a wide array of bacteria and fungi produce organic acids as natural chelating agents of HM. Acidic pH conditions favor bioavailability and adsorption of HM in the rhizosphere. Similarly, the metal-removal efficiency of *Trichoderma* sp. can be enhanced with BC application as the latter not only provides energy to the *Trichoderma* sp. but also performs similar chelating mechanisms to remove HMs from soil [19]. Herliana [22] stated that the application of *T. harzianum* showed the higher adsorption of Pb in soil by intracellular metal immobilization and biosorption of metal ions to cell wall of fungi.

### 3.4. Impact of Biochar and T. asperellum on Maize ROS and Antioxidant Enzymes under HM Stresses

The Cu and Ni stresses in the rhizosphere cause the production of ROS such as H_2_O_2_ which disrupts the cellular homeostasis causing oxidative damage in both cultivars irrespective of the applied treatments (Figure 6). It leads to disturb the plant biochemical mechanisms by producing MDA fragments cause lipid peroxidation [3]. Different studies reported the elevated concentrations of MDA in maize and rice under Cr and Cd toxicity, respectively [25,47]; however, BC and *T. asperellum* application reduced the impact of Cu and Ni stress on maize plant by significantly reducing the H_2_O_2_ and MDA contents (Figure 6). The plant physiology is positively regulated by the application of *Trichoderma harzianum* against oxidative stress damage and mineral uptake by significantly enhancing the hormonal efficiency and development of antioxidant metabolism, as reported by Ahmad [48]. *Trichoderma harzianum* alleviated the oxidative stress altering the phytohormone levels and phosphate solubilization ability [49]. Another study by Yaghoubian [16] investigated the effect of *T. asperellum* application and reported the reduced concentration of MDA (39%) and H_2_O_2_ (42%) by stimulating the activity of antioxidant enzymes in *Spinacia oleracea* L. plant.

In our study, there was significant genotypic variations among both cultivars shown for antioxidant enzymes activities with BC and fungal application under Cu and Ni stress (Figure 7 and Figure 8). In the study of Naeem et al. [50], Si application significantly improved the CAT and APX activities under Cd stress and Si induced improvement was higher for Iqbal-2000 (67 and 52%, respectively) than Sehar-2006 (51 and 47%, respectively) wheat cultivar. Additionally, the content of H_2_O_2_ was also estimated to reduce under coupled application of treatment (BC+ *T. asp*) and this might be attributed to the conversion of H_2_O_2_ to water and molecular oxygen by the antioxidative enzymes (APX and CAT) (Figure 6). For estimating the steady-state amount of ROS excess, the equilibrium between these enzymes is critical. APX and CAT have different affinity for H_2_O_2_, indicating that they relate to two separate groups of H_2_O_2_-scavenging enzymes. Our results are consistent with the study of Qi and Zhao [28], which showed that under strontium stress, SOD activity of oat (Neimengkeyi-1) tolerant genotype was enhanced up to 2-fold in both roots and leaves as compared to (Bayou-3) sensitive genotype. Similarly, in our study, SOD activity of spring corn leaf was enhanced by (3-fold) as whereas in footer corn SOD activity in leaf was enhanced by (2-fold) under Cu toxicity. This implies that spring corn is a HM tolerant cultivar as compared to footer corn (sensitive) cultivar, which can be further tested on a field scale for cultivation.

## 4. Materials and Methods

### 4.1. Preparation and Handling of Microbial Isolates

Pure cultures of fungal species *Trichoderma harzianum* (FCBP-SF-1277) (F1), *Trichoderma asperellum* (FCBP-SF-1290) (F2) and *Trichoderma viride* (FCBP-SF-639) (F3) were acquired from the first fungal culture bank of Pakistan (FCBP) at the Faculty of Agricultural Sciences, University of the Punjab, Lahore. Prior to making media for fungal isolates, the entire apparatus was sterilized firstly with detergent/ ethanol and finally autoclaved (KT- 30L) at 121 °C for 15 to 20 min. A laminar air flow cabinet (Technical Safety Services LLC. Berkely, CA, USA) was used for pouring of media into petri dishes, its solidification and the inoculation of fungal isolates. Prior to use, laminar air flow was sterilized with ethanol, UV light was turned on for 15 to 20 min, following precautionary measures, and flame was also turned on prior to work to avoid any kind of contamination [51].

Potato dextrose media was used for cultivation and identification of fungi. There were two types of nutrient media: (a) potato dextrose agar (PDA) and (b) potato dextrose broth (PDB). To prepare PDA media: firstly, in 500 mL distilled water, 19.5 g commercial PDA powder was dissolved. Using NaOH, the pH of the solution was adjusted up to pH 5.6 ± 0.2 at 25 °C followed by autoclave for 15 min at 15 psi, 121 °C, then cooled down to 55 °C. An antibacterial (Streptomycin) capsule was added into this solution and the solution poured into petri dishes, and their lids were tightly closed. Required time was given for the solution to harden, and it was then stored in the dark at 4 °C to assure any contamination was avoided. After isolation and purification, fungal isolates were characterized by means of their morphology [52]. Fungal cultures were preserved at 4 °C for further use. To prepare broth media, 200 g potato cubes were boiled in 1000 mL distilled water till 500 mL water was left. No agar was added, to avoid solidification of media in 1000 mL conical flasks [51].

### 4.2. Lab Trial (Study 1)

#### 4.2.1. Biosorption Trial of Fungal Isolates 

In this experiment, fungal isolates (F1, F2 and F3) were screened for their tolerance against Cu (100, 200, 300 mg/Kg), Ni (50, 100, 150 mg/Kg) and Pb (50, 100, 150 mg/Kg) individually in PDB. There were in total 20 treatments for each fungal isolate. Each treatment was replicated three times. All the fungal strains were inoculated in PDB medium containing respective concentrations of HMs separately. Inoculation of fungal isolates in normal PDB serves as control for the comparison of growth of fungal isolates in PDB medium. Conical flasks were kept in a shaker incubator at 28 °C ± 2 °C and 200 rpm. The growth of fungal biomass was observed after the 5th and 7th day. Followed by filtration (Whatman No. 42) and centrifugation at 1500× *g* for 5 min, the debris and the supernatant (without fungal particles) were collected separately and used to analyze residual concentration of HMs using a multisequential atomic absorption spectrophotometer (AAS) (iCE 3000 SERIES) (Thermo Fisher Scientific, Waltham, MA, USA). The initial and final metal concentrations were compared using the statistical analysis.

#### 4.2.2. Removal Efficiency %

Removal efficiency of the studied fungal isolates were determined using the Equilibrium (1) presented by Mohsenzadeh and Shahrokhi [32].
(1)$$R=\left(\frac{P_{0}-P_{e}}{P_{0}}\right)\times 100$$

In this equation, *R* = the percentage of metal removal by the fungal biomass, *P*0 = the initial concentration of metal ions (ppm) and *Pe* = the final concentration of metal ions (ppm) in the experimental media.

#### 4.2.3. HM Removal Efficiency (q) Per Unit Mass of Fungi

To analyze the dry weight of the fungal biomass, fresh fungal biomass was placed in the oven at 105 °C for 48 h. The equilibrium (2) was used later presented by Mohsenzadeh and Shahrokhi [32].
(2)$$q=\left(\frac{C_{i}-C_{e}}{M}\right)\times V$$

In this equation, *q* = the metal absorbance based on dry weight of fungi (mg/g), *Ci* = the initial concentration of metal at the beginning of experiment (ppm), *Ce* = the metal concentration at the end of experiment (ppm), *V* = the volume of the solution (l) and *M* = fungal dry mass (g).

### 4.3. Preparation of Rice Straw Biochar

The rice straw biochar (BC) was prepared by following the protocol of Khan [53]. The prepared BC was characterized through X-ray fluorescence, scanning electron microscope and Fourier Transform Infrared Spectroscopy; the images were published in our earlier paper [53].

### 4.4. Greenhouse Trial (Study 2)

The proposed study was conducted in a greenhouse at the Botanic Garden, Government College University Lahore at 31°33′41.94″ E latitude and 74°19′41.94″ N longitude due to specific concerns regarding safety and management. According to the Pakistan Metrological Department, the average maximum and minimum temperature is 42 °C and 25 °C, respectively, and the annual rainfall is about 626 mm at the region of study site. Bhal soil was used for the experiment and characterized as [Soil texture (clay loam), Sand (37%), Silt (34%), Clay (35%), organic matter (0.88%) [54,55], pH (7.45) [56], Electrical conductivity (1.36 d·Sm^−1^) [57], Cation exchange capacity (5.93 cmol Kg^−1^) (Hailegnaw et al. 2019), Extractable Cu (0.4018 mg Kg^−1^) and Extractable Ni (0.3873 mg Kg^−1^)] in the laboratory [58].

Two maize cultivars “spring corn” (V1) and “footer corn” (V2) were selected for the experiment. There were in total 11 treatments: T1 = Control, T2 = Cu (300 ppm), T3 = Ni (100 ppm), T4 = *T. asp*, T5 = BC (2%), T6 = *T. asperellum* + Cu (300 ppm), T7 = *T. asperellum* + Ni (100 ppm), T8 = BC (2%) + Cu (300 ppm), T9 = BC (2%) + Ni (100 ppm), T10 = *T. asperellum* + BC (2%) + Cu (300 ppm) and T11 = *T. asperellum* + BC (2%) + Ni(100 ppm) applied in the pots, each with three kg of soil. Each treatment was replicated three times and an adequate amount of water was applied to all pots to maintain soil field capacity.

A conidial suspension was prepared using a scraping method in sterile distilled water, as fungal cultures could not be applied directly into soil. For this purpose, the fungal mycelium was collected by adding 5 mL autoclaved water to each Petri dish and the surface was scraped with a spatula. The conidia number per mL was adjusted to 1.0 × 10^7^ conidia mL^−1^ using a Neubauer haemocytometer. This dilution was added in soil @ 10% per 1 Kg of soil to carry out the pot experiment. Different proportions of treatments were added into the pots for both cultivars. 

After the 2 days of applying treatments, 6–7 seeds of maize were sown in each pot. After 8 days of sowing, Urea and DAP were added in the pots, respectively. After 3 months, the plants were harvested; roots and shoots of plants were carefully extracted and washed with distilled water to remove unwanted materials. Plants and soil samples were stored in appropriately labelled zipper bags. Then, soil and plants were carried into the laboratory for further analysis.

### 4.5. Measurements and Analysis

#### 4.5.1. Maize Physiological Characteristics

The length and fresh weight of roots and shoots were recorded using standard measuring tape and a weighing balance. To analyze dry weight, fresh plants were first dried in open air for 2 days followed by in a hot oven at 65 °C for 24 h. The proposed method of Du [59] was used to calculate the chlorophyll content. For this purpose, 1 g of fresh maize leaves were ground with 80% acetone and centrifuged (HERMLE Z167M) at 4000× *g* for 5 min. Absorbance was recorded using UV/visible spectrophotometer (Shimadzu UV-1201, Kyoto, Japan) at 663 nm and 645 nm.

#### 4.5.2. Maize Biochemical Characteristics 

The concentrations of protein in root, stem and leaves of maize plants were determined using the method of Metwally [60]. A total of 0.1 g of fresh maize root, stem and leaves were homogenized in 80% acetone to prepare sample extract. A reaction mixture containing sample extract (200 µL), d. H_2_O (1800 µL) and 2 mL of Bradford reagent was prepared and incubated at room temperature for 10–20 min. After incubation, absorbance was recorded using a UV/visible spectrophotometer (Shimadzu UV-1201, Kyoto, Japan) at 595 nm.

The concentrations of total phenolics in root, stem and leaves of maize plants were determined using the method of Ahmed [61]. A reaction mixture containing sample extract (20 µL), d. H_2_O (1580 µL), Na_2_CO_3_ (300 µL) and 100 µL of Folin-Coicâlteu’s reagent was prepared and incubated in the dark at room temperature for 2 h. A spectrophotometer was used to measure the absorbance at 760 nm after incubation.

#### 4.5.3. Estimation of Cu and Ni in Plants and Soil

The Di-Acid digestion method was used to analyse the concentrations of HMs such as Cu and Ni in maize and soil. Concentrations of HMs were analysed using a multi-sequential AAS (iCE 3000 SERIES) against prepared standards of known concentration [62].

#### 4.5.4. Estimation of Maize Malondialdehyde and Hydrogen Peroxide Concentration

MDA concentrations were analysed using the protocol of Kanwal [36]. For this purpose, 0.5 g of fresh maize samples were homogenized in 5 mL of 20% TCA solution, followed by centrifugation at 10,000× *g* for 15 min. In a dry oven, a reaction mixture containing sample extract, TCA and TBA (thiobarbituric acid) was heated for 30 min at 95 °C and then cooled at freezing temperature. The absorbance was recorded at 532 nm and 600 nm using a UV/visible spectrophotometer (Shimadzu UV-1201, Kyoto, Japan). The concentration of MDA was determined using Beer and Lambert’s equation [63].

To determine H_2_O_2,_ fresh maize samples (0.5 g) of each plant part were ground with 5 mL of Trichloroacetic acid (TCA) (0.1%) solution [64]. The mixture was centrifuged at 12,000× *g* for 15 min. After centrifugation, a reaction mixture was prepared using 1 mL of supernatant, 1 mL 10 mM K-P buffer (pH 7.0) and 2 mL 1M potassium iodide (KI) solution. The concentration of H_2_O_2_ was calculated by recording the absorbance of the mixture at 390 nm.

#### 4.5.5. Estimation of Antioxidant Enzyme Activity

To calculate the concentration of Superoxide Dismutase (SOD), an Ascorbate peroxidase (APX), Catalase (CAT) and Guaiacol peroxidase (POD) enzyme assay was prepared using 1 g of fresh root, stem and leaf in potassium phosphate buffer followed by centrifugation at 15,000× *g* at 4 °C for 20 min. Supernatant (sample extract) was used for further analysis [65].

For SOD, a reaction mixture was prepared using sample extract (70 μL), 1630 μL of sodium carbonate buffer (pH = 10.2), 500 μL of nitroblue tetrazolium (24 μM), 100 μL of ethylenediamine tetraacetic acid (EDTA) (0.1 mM), 100 μL of hydroxylamine hydrochloride (1 mM) and 100 μL of Triton-X-100 (0.03%). To express the activity of SOD, absorbance was recorded at 560 nm using a spectrophotometer (Shimadzu UV-1201, Kyoto, Japan) [66].

For APX, a reaction mixture was prepared containing sample extract, 50 mM K-P buffer (pH 7.0), 0.5 mM Ascorbate and 0.5 mM H_2_O_2_. To express the activity of APX, absorbance was recorded at 290 nm using a spectrophotometer (Shimadzu UV-1201, Kyoto, Japan) [67]. To calculate the CAT activity, the absorbance of a reaction mixture containing sample extract was observed at 240 nm [65]. For POD, a reaction mixture was prepared using sample extract (70 μL), 2350 μL potassium phosphate buffer (K-P buffer) (50 mM, pH 7.0), 50 μL guaiacol solution (20 mM) and 30 μL H_2_O_2_ (12 mM). To express the activity of POD, absorbance was recorded at 436 nm using a spectrophotometer (Shimadzu UV-1201, Kyoto, Japan) [66].

#### 4.5.6. Physiochemical Analysis of Soil

Soil pH was measured by forming a soil suspension of 50 g soil sample in 50 mL d.H_2_O following the protocol of Hailegnaw [57]. After 1 h of filtering, the suspension and EC of clear soil extract was measured using an EC meter [57]. Soil moisture (%) was analysed by the following formula. The water-holding capacity of soil was analysed by taking 100 g of air-dried soil and gradually adding water until the soil was covered. The amount of added water was carefully recorded. The mix was stirred and allowed to sit. After 1-h suspension, the mix was filtered through Whatman filter paper. How much water was retained in the 100 g of soil sample was calculated using the formula
(3)$$Soil\;Moisture\;\%=\frac{Wet\;weight-Dry\;weight*100}{Dry\;weight}\;$$
(4)$$\;\;\frac{Water\;retained\;\mathrm{mL}}{100\;\mathrm{mL}\;sample}=water\;added\;\left(\mathrm{mL}\right)-water\;drained\;\left(\mathrm{mL}\right)$$
(5)$$\mathrm{Water}\;\mathrm{Holding}\;\mathrm{Capacity}\;(\mathrm{mL}/L)=\frac{Water\;retained\;\mathrm{mL}}{100\;\mathrm{mL}\;sample}\;\text{×}\;10\;$$

### 4.6. Statistical Analysis

Microsoft Excel 2019^®^ and Statistix 8.1^®^ (Analytical software, Tallahassee, FL, USA) tools were used to analyze the data. A two-factor factorial design (analysis of variance) was used to statistically analyze the datasets of the above-mentioned treatments. To compare the mean groups, the Least Significant Difference (LSD) test was applied through Statistix 8.1^®^ [68]. To find and evaluate the best treatment among different parameters of root, stem, leaves and soil, principal component analysis (PCA-Pearson correlation) was performed using XLSTAT version 2021 [67]. The results showed the mean of three replicates (*n* = 3) with standard error ± (*SE)*.

## 5. Conclusions

There was significant genotypic variation shown by both maize cultivars, and spring corn proved to be a tolerant cultivar, compared to footer corn, against applied HM stress. *T. asperellum* species were screened out initially from the biosorption trial on the basis of their performance against Cu and Ni stresses; latterly, *T. asperellum* species with BC provided significant results by improving physiological and biochemical parameters of maize cultivars against Cu stress. Hence, applications of BC with interaction of fungi proved to be a cost-effective and eco-friendly soil conditioner in developing countries including Pakistan to limit the deleterious effects of HM toxicity.

## Figures and Tables

**Figure 1 plants-12-00958-f001:**
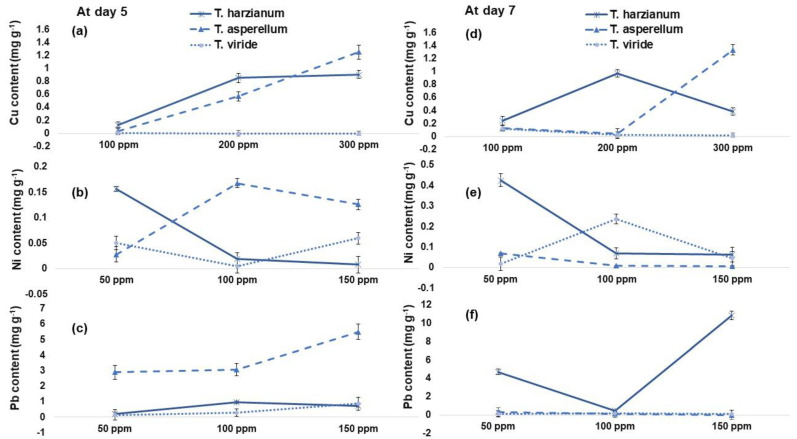
(**a**–**c**) Total copper (Cu), nickel (Ni) and lead (Pb) content in *Trichoderma harzianum* (F1), *Trichoderma asperellum* (F2) and *Trichoderma viride* (F3) at day 5. (**d**–**f**) Total copper (Cu), nickel (Ni) and lead (Pb) content in *Trichoderma harzianum* (F1), *Trichoderma asperellum* (F2) and *Trichoderma viride* (F3) at day 7, respectively. Values are the average of three replicates.

**Figure 2 plants-12-00958-f002:**
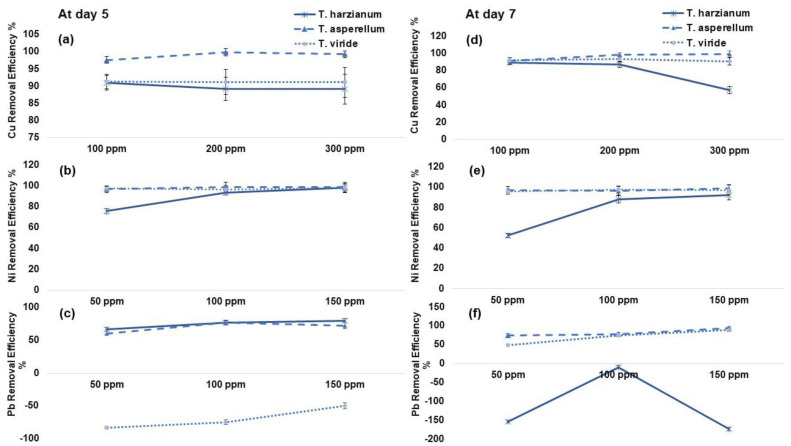
(**a**–**c**) Removal efficiency % of copper (Cu), nickel (Ni) and lead (Pb) by *Trichoderma harzianum* (F1), *Trichoderma asperellum* (F2) and *Trichoderma viride* (F3) at day 5. (**d**–**f**) Removal efficiency % of copper (Cu), nickel (Ni) and lead (Pb) by *Trichoderma harzianum* (F1), *Trichoderma asperellum* (F2) and *Trichoderma viride* (F3) at day 7, respectively. Values are the average of three replicates.

**Figure 3 plants-12-00958-f003:**
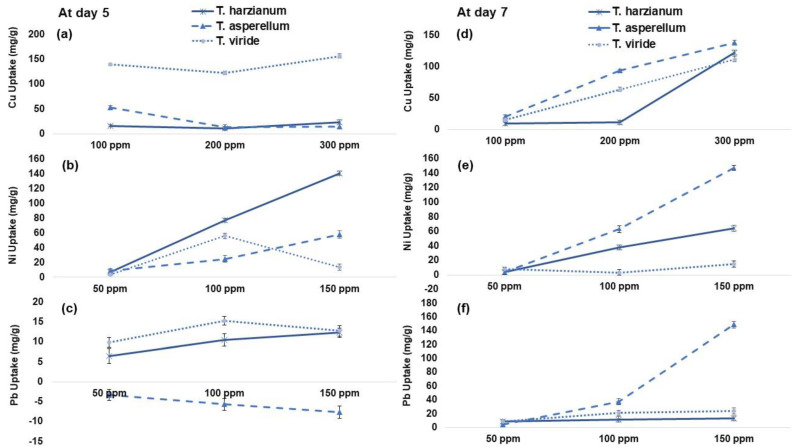
(**a**–**c**) Total copper (Cu), nickel (Ni) and lead (Pb) uptake per gram dry biomass of *Trichoderma harzianum* (F1), *Trichoderma asperellum* (F2) and *Trichoderma viride* (F3) at day 5. (**d**–**f**) Total copper (Cu), nickel (Ni) and lead (Pb) uptake per gram dry biomass of *Trichoderma harzianum* (F1), *Trichoderma asperellum* (F2) and *Trichoderma viride* (F3) at day 7, respectively. Values are the average of three replicates.

**Figure 4 plants-12-00958-f004:**
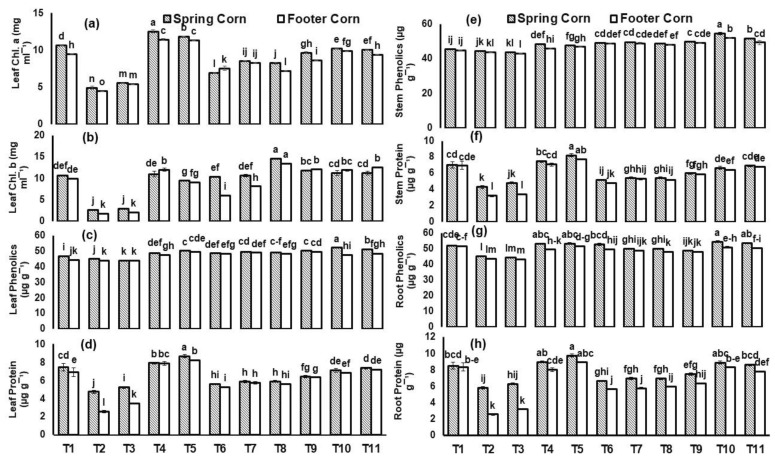
(**a**) Chlorophyll a contents in leaf, (**b**) Chlorophyll b contents in leaf, (**c**,**d**) Leaf phenolics and proteins, (**e**,**f**) Stem phenolics and proteins and (**g**,**h**) Root phenolics and proteins of *Zea mays* L. plant grown in heavy-metal-contaminated soil under different treatments such as T1 = Control; T2 = Cu(300 ppm); T3 = Ni(100 ppm); T4 = F2; T5 = BC(2%); T6 = F2 + Cu(300 ppm); T7= F2 + Ni(100 ppm); T8 = BC(2%) + Cu(300 ppm); T9 = BC(2%) + Ni(100 ppm); T10 = F2+ BC(2%) + Cu(300 ppm) and T11= F2+ BC(2%) + Ni(100 ppm). Values are the average of three replicates ± standard error (SE). Mean bars sharing similar lower-case alphabetic letters were non-significant (*p* < 0.05) to each other, while different letters represented significant difference declared by one-way ANOVA using LSD test at *p* ≤ 0.05.

**Figure 5 plants-12-00958-f005:**
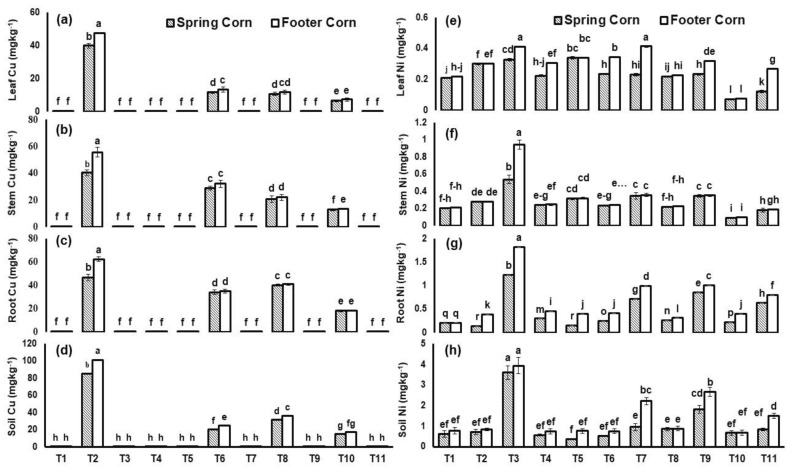
Concentration of total copper (Cu) in (**a**) leaf, (**b**) stem, (**c**) root, (**d**) soil and nickel (Ni) in (**e**) leaf, (**f**) stem, (**g**) root and (**h**) soil of *Zea mays* L. plants grown in heavy-metal-contaminated soil under different treatments such as T1 = Control; T2 = Cu(300 ppm); T3 = Ni(100 ppm); T4 = F2; T5 = BC(2%); T6 = F2 + Cu(300 ppm); T7= F2 + Ni(100 ppm); T8 = BC(2%) + Cu(300 ppm); T9 = BC(2%) + Ni(100 ppm); T10 = F2+ BC(2%) + Cu(300 ppm) and T11= F2+ BC(2%) + Ni(100 ppm). Values are the average of three replicates ± standard error (SE). Mean bars sharing similar lower-case alphabetic letters were non-significant (*p* < 0.05) to each other while different letters represented significant difference declared by one-way ANOVA using LSD test at *p* ≤ 0.05.

**Figure 6 plants-12-00958-f006:**
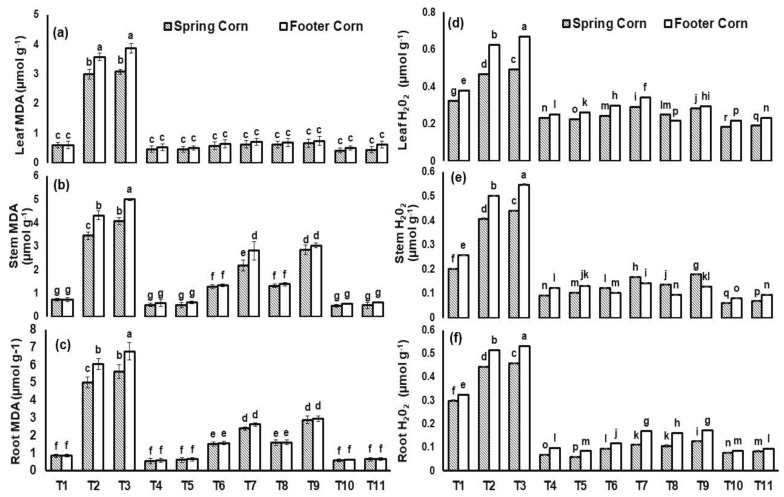
Malondialdehyde (MDA- µmol g^−1^) in (**a**) leaf, (**b**) stem and (**c**) root and hydrogen peroxide (H_2_O_2_- µmol g^−1^) in (**d**) leaf, (**e**) stem and (**f**) root of *Zea mays* L. plant grown in heavy-metal-contaminated soil under different treatments such as T1 = Control; T2 = Cu (300 ppm); T3 = Ni (100 ppm); T4 = F2; T5 = BC (2%); T6 = F2 + Cu (300 ppm); T7 = F2 + Ni (100 ppm); T8 = BC (2%) + Cu (300 ppm); T9 = BC (2%) + Ni (100 ppm); T10 = F2 + BC (2%) + Cu (300 ppm) and T11 = F2 + BC (2%) + Ni (100 ppm). Values are the average of three replicates ± standard error (SE). Mean bars sharing similar lower-case alphabetic letters were non-significant (*p* < 0.05) to each other while different letters represented significant difference declared by one-way ANOVA using LSD test at *p* ≤ 0.05.

**Figure 7 plants-12-00958-f007:**
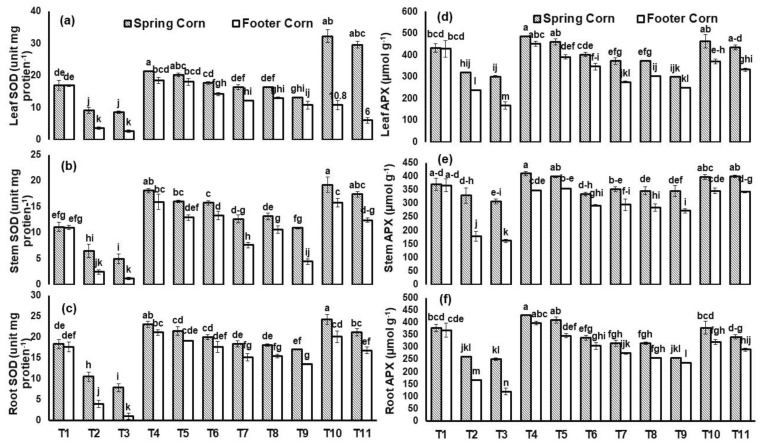
Superoxide dismutase (SOD- (unit mg protein^−1^) in (**a**) leaf, (**b**) stem and (**c**) root and ascorbate peroxidase (APX- (µmol g^−1^) in (**d**) leaf, (**e**) stem and (**f**) root of *Zea mays* L. plant grown in heavy-metal-contaminated soil under different treatments such as T1 = Control; T2 = Cu (300 ppm); T3 = Ni (100 ppm); T4 = F2; T5 = BC (2%); T6 = F2 + Cu (300 ppm); T7 = F2 + Ni (100 ppm); T8 = BC (2%) + Cu (300 ppm); T9 = BC (2%) + Ni (100 ppm); T10 = F2 + BC (2%) + Cu (300 ppm) and T11 = F2 + BC (2%) + Ni (100 ppm). Values are the average of three replicates ± standard error (SE). Mean bars sharing similar lower-case alphabetic letters were non-significant (*p* < 0.05) to each other while different letters represented significant difference declared by one-way ANOVA using LSD test at *p* ≤ 0.05.

**Figure 8 plants-12-00958-f008:**
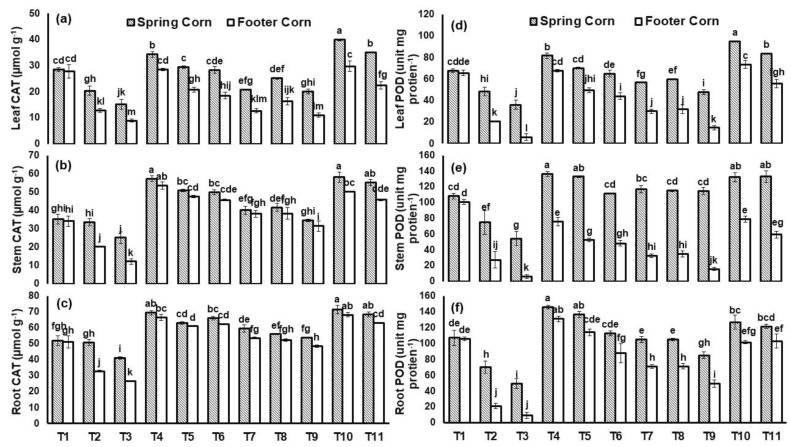
Catalase (CAT- (µmol g^−1^) in (**a**) leaf, (**b**) stem and (**c**) root and guaiacol peroxidase (POD- (unit mg protein^−1^) in (**d**) leaf, (**e**) stem and (**f**) root of *Zea mays* L. plant grown in heavy-metal-contaminated soil under different treatments such as T1 = Control; T2 = Cu (300 ppm); T3 = Ni (100 ppm); T4 = F2; T5 = BC (2%); T6 = F2 + Cu (300 ppm); T7= F2 + Ni (100 ppm); T8 = BC (2%) + Cu (300 ppm); T9 = BC (2%) + Ni (100 ppm); T10 = F2 + BC (2%) + Cu (300 ppm) and T11= F2 + BC (2%) + Ni (100 ppm). Values are the average of three replicates ± standard error (SE). Mean bars sharing similar lower-case alphabetic letters were non-significant (*p* < 0.05) to each other while different letters represented significant difference declared by one-way ANOVA using LSD test at *p* ≤ 0.05.

**Figure 9 plants-12-00958-f009:**
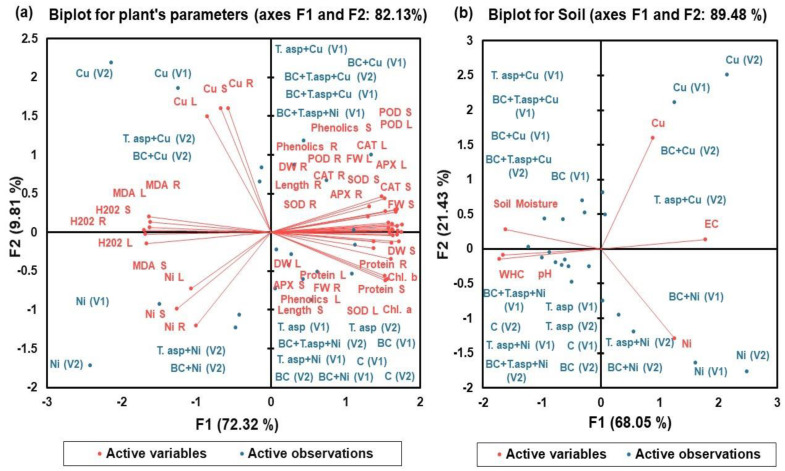
Principal component analysis (PCA) of two maize cultivars “spring corn” (V1) and “footer corn” (V2) and soil parameters. The two main factors of PCA (F1 and F2) significantly contributed to generate PCA biplots. Active variables are (**a**) Length, Fresh weight (FW), Dry weight (DW), Chlorophyll a (Chl. a), Chlorophyll b (Chl. b), Proteins, Phenolics, Hydrogen peroxide (H_2_O_2_), Malondialdehyde (MDA), Ascorbate peroxidase (APX), Catalase (CAT), Superoxide Dismutase (SOD), Guaiacol peroxidase (POD), total Cu and total Ni and (**b**) parameters of soil such as Ph, EC, Soil Moisture%, Water Holding Capacity (WHC), total Cu and total Ni; however, R, S and L stand for root, stem and leaf, respectively. Active observations are control, Copper (Cu), Nickel (Ni), *Trichoderma asperellum (T. asp)*, simple Biochar (BC).

**Table 1 plants-12-00958-t001:** Growth parameters of two maize cultivars were determined under different treatments against copper and nickel stress.

Cultivars		Spring Corn											Footer Corn										
Treatments		T1	T2	T3	T4	T5	T6	T7	T8	T9	T10	T11	T1	T2	T3	T4	T5	T6	T7	T8	T9	T10	T11
Length (cm)	Shoot	30 ± 1 gh	22 ± 0.5 k	20 ± 0.3 lm	34 ± 0.5 bc	35 ± 0.3 a	33 ± 0.3 cd	35 ± 0.3 cd	31 ± 0.3 fg	28 ± 0.2 i	33 ± 0.3 de	34 ± 0.3 bc	28 ± 0.5 i	21 ± 0.5 kl	19 ± 0.3 m	34 ± 0.5 bc	33 ± 0.2 bcd	31 ± 0.3 ef	33 ± 0.3 cd	29 ± 0.5 h	26 ± 0.3 j	32 ± 0.3 de	33 ± 0.2 bc
	Root	11 ± 0.3 g	6 ± 0.3 i	4 ± 0.2 j	15 ± 0.4 d	17 ± 0.2 c	12 ± 0.4 f	11 ± 0.4 g	10 ± 0.3 h	9 ± 0.5 h	20 ± 0.6 a	18 ± 0.3 b	9 ± 0.1 h	5 ± 0.4 j	3 ± 0.3 j	13 ± 0.4 f	14 ± 0.3 e	10 ± 0.2 g	9 ± 0.3 h	8 ± 0.3 h	6 ± 0.3 i	16 ± 0.7 c	14 ± 0.6 de
F.W (g)	Shoot	11 ± 0.2 i	9 ± 0.2 j	7 ± 0.5 k	16.6 ± 0.4 cde	17 ± 0.2 bcd	15.5 ± 0.6 fg	14.5 ± 0.4 g	15.5 ± 0.3 de	14.3± 0.4 fg	19 ± 0.3 a	17 ± 0.5 b	9 ± 4 i	7 ± 2.1 k	5 ± 1.1 l	14 ± 2.1 fg	15.5 ± 3.6 ef	13.4 ± 4.9 g	12 ± 4.6 h	14 ± 5.0 fg	14 ± 4.4 g	17 ± 5.4 bc	15 ± 5.2 ef
	Root	4 ± 0.2 d	0.6 ± 0.1 g	0.5 ± 0.1 gh	4.5 ± 0.1 b	4 ± 0.2 b	4 ± 0.1 bc	3.6 ± 0.1 d	3.7 ± 0.1 d	3 ± 0.1 ef	5 ± 0.1 a	4 ± 0.1 b	3 ± 0.1 ef	0.4 ± 0.1 gh	0.3 ± 0.2 h	4 ± 0.1 b	3.7 ± 0.3 d	3 ± 0.2 e	3 ± 0.1 ef	3 ± 0.2 ef	2 ± 0.2 f	4 ± 0.3 b	3.5 ± 0.2 cd
D.W (g)	Shoot	1.8 ± 0.02 fg	0.63 ± 0.02 h	0.4 ± 0.02 h	2.8 ± 0.1 cd	3.2 ± 0.2 ab	2.5 ± 0.1 e	2.2 ± 0.3 e	2.6 ± 0.3 bc	2.3 ± 0.2 cd	3.6 ± 0.1 a	2.9 ± 0.5 ab	1.5 ± 0.01 h	0.41 ± 0.01 h	0.3 ± 0.01 h	2.1 ± 0.5 e	2.3 ± 0.2 e	1.8 ± 0.4 f	1.5 ± 0.4 g	2 ± 0.1 e	1.8 ± 0.5 e	2.5 ± 0.3 d	2 ± 0.2 e
	Root	0.8 ± 0.1 gh	0.7 ± 0.1 ijk	0.6 ± 0.1 jk	1.2 ± 0.3 cd	1.3 ± 0.5 bc	1.2 ± 0.5 de	1 ± 0.2 ef	1.1 ± 0.4 de	1 ± 0.1 de	1.5 ± 0.3 ab	1.4 ± 0.2 a	0.7 ± 0.1 gh	0.5 ± 0.1 k	0.4 ± 0.1 k	0.9 ± 0.1 fg	1 ± 0.1 ef	0.8 ± 0.1 ghi	0.7 ± 0.1 hij	1 ± 0.5 ef	0.9 ± 0.1 fg	1.3 ± 0.4 ab	1 ± 0.5 cd

Data in Table 1 represents the mean values of three replicates (*n* = 3) ± standard error, and different letters represent significant differences declared by the LSD test at *p* ≤ 0.05. Cu-Copper 300 ppm; Ni-Nickel 100 ppm; *T. asp-Trichoderma asperellum*; BC-rice husk bioch.

## Data Availability

Data has been collected massively from different research sources and can be tracked through given reference links.

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
