# Peer review of "Trichoderma asperellum L. Coupled the Effects of Biochar to Enhance the Growth and Physiology of Contrasting Maize Cultivars under Copper and Nickel Stresses"

_plants, 2023, doi:10.3390/plants12040958_

Round 1

Reviewer 1 Report

This Mss pprovides useful information and should be publsihed. I have some concerns however.

First, I have never heard of footer corn. Can the authors define this?

Line 29--please substite plant weight for body weight..

Line 73--3% of what?

Fig. 2--Would it be more straight-forward to provide data as PPM accumulation rather than the calculated efficency? At the least, please provide the e\definition of the terms in the equations on page 4.

On line 125 and elsewhere, "folds" should be "fold"

Please spell out asperellum rather than asp.

Tables 1, 2 and 3 are long with many data points. I wonder if the authors would be better served by choosing fewer stages of growth. I am concerned that the length of the tables will reduce impact on the readers.

Line 283--please omit "burst:

Line 331-333--italicize Trichoderma

Author Response

Comment 1: I have never heard of footer corn. Can the authors define this?

Response: Thank you for the comment. Footer Corn is the local maize genotype in Pakistan with Code (FG2002).

Comment 2: Line 29--please substitute plant weight for body weight.
Response: Comment was addressed. Please see Line 30-32

Comment 3: Line 73--3% of what?
Response: Comment was addressed. Please see line 75-76.

Comment 4: Fig. 2--Would it be more straight-forward to provide data as PPM accumulation rather than the calculated efficiency? At the least, please provide the e\definition of the terms in the equations on page 4.

Response: Thank you for the comment. The e\definition of the terms in the equations on page 10 has been added. Many scientists already mentioned removal efficiency in their research work such as:

Mohsenzadeh F, Shahrokhi F (2014) Biological removing of Cadmium from contaminated media by fungal biomass of Trichoderma species. J Environ Health Sci and Eng, 12(1), 1-7. https://doi.org/10.1186/2052-336X-12-102

Comment 5: On line 125 and elsewhere, "folds" should be "fold"
Response: Comment has been addressed. Please see line 106-107. Whole manuscript has been re-evaluated as directed by the reviewer.

Comment 6: Please spell out asperellum rather than asp.
Response: Comment has been addressed. Whole manuscript has been re-evaluated as directed by the reviewer.

Comment 7: Tables 1, 2 and 3 are long with many data points. I wonder if the authors would be better served by choosing fewer stages of growth. I am concerned that the length of the tables will reduce impact on the readers.
Response: Thank you for the comment. In table 1, fewer growth stages have been chosen to summarize the data. In case of Table 2, 3 and 4, we converted the data to figures 4, 5, 7 and 8 to facilitate the readers.

Comment 8: Line 283--please omit "burst:

Response: Comment has been addressed. Please see line 341.

Comment 9: Line 331-333--italicize Trichoderma
Response: Comment was addressed. Please see line 390-392. Whole manuscript has been re-evaluated as well.

Reviewer 2 Report

The manuscript covers issues that are certainly current and of interest. The abstract and introductory section are well written. The results section is a bit difficult to follow, while the discussion does not discuss the results obtained in detail, but seems more like a review of the bibliography that showed results in line with what the authors of this paper found.

The tables and figures are very poorly edited. More care and attention would be appropriate when a scientific paper is submitted to a journal for publication.

In general, the novelty of this study does not emerge in the various sections of the manuscript.

I recommend that the authors carefully review the manuscript considering the previous comments.

Secondary issues 

Line 25: I would not call HMs as byproducts, they are contaminants. 

Line 52: detox in parentheses does not seem to render the meaning ascribed to glutathione in deoxification. It would be better to remove that parenthesis or better explain the meaning. 

 Line 69: missing the name referred to in the reference

Line 73: what is meant by reducing electron leakage? 

Line 76: the same as in line 69

Figure 1: the standard deviation is missing. Same as in figure 2 and 3. 

Table 1 in the present form is unreadable. It needs to be improved and edited much more. The same applies to Tables 2, 3 and 4. 

Author Response

Comment 1: The results section is a bit difficult to follow.
Response: Comment was addressed. Please check Line 149-157 and  201-205.

Comment 2: Discussion does not discuss the results obtained in detail, but seems more like a review of the bibliography that showed results in line with what the authors of this paper found.

Response: Thank you for the comment. The authors have improved the whole discussionby improving its structure and coherrence. 4 subheadings has also been added to rationalize the topic. Please see page 6-8. We are happy to improve it further, if reviewer suggest us in next round of revision.

Comment 3: Line 25: I would not call HMs as byproducts, they are contaminants. Response: Comment was addressed. Please see Line 28.

Comment 4: Line 52: detox in parentheses does not seem to render the meaning ascribed to glutathione in detoxification. It would be better to remove that parenthesis or better explain the meaning.
Response: Comment was addressed. Please see Line 54-56.

Comment 5:  Line 69: missing the name referred to in the reference
Response: Thank you for the comment. In line 70-72 the name “NEELUM” is referred to the maize cultivar and not to an author’s name.

Comment 6: Line 73: what is meant by reducing electron leakage? 
Response: Comment was addressed. Please see Line 76-79.

Comment 7: Line 76: the same as in line 69
Response: Thank you for the comment. In line 78, the name Kasur and Sialkot are referred to the cities of Pakistan and not to an author’s name.

Comment 8: Figure 1: the standard deviation is missing. Same as in figure 2 and 3. 
Response: Comment has been addressed. Required Standard deviation bars has been added to figure 1, 2 and 3.

Comment 9: Table 1 in the present form is unreadable. It needs to be improved and edited much more. The same applies to Tables 2, 3 and 4. 
Response: Thankyou for the comment. In table 1, fewer growth stages have been chosen to summarize the data. In case of Table 2, 3 and 4, we convert the data to figures 4, 5, 7 and 8 to facilitate the readers.

Round 2

Reviewer 2 Report

The manuscript, after the revision done by the authors, can be accepted for publication.